# UICopilot: Automating UI Synthesis via Hierarchical Code Generation from Webpage Designs

## Abstract

Automating the synthesis of *User Interfaces* (UIs) plays a crucial role in enhancing productivity and accelerating the development lifecycle, reducing both time and manual effort. Recently, the rapid development of *Multimodal Large Language Model* (MLLM) has made it possible to generate front-end *Hypertext Markup Language* (HTML) code directly from webpage designs. However, real-world webpages encompass not only a diverse array of HTML tags but also complex stylesheets, resulting in significantly lengthy code. The lengthy code challenges the performance and efficiency of MLLMs, especially in capturing UI's structure information. To mitigate this challenge, this paper puts forward UICopilot, a novel approach to automating UI synthesis via hierarchical code generation from webpage designs. The core idea of UICopilot is to decouple the process into two stages: generating the coarse HTML hierarchical structure and then producing fine-grained code. To validate the effectiveness of UICopilot, we conduct experiments on a real-word dataset, i.e., Vision2UI. Experimental results demonstrate that UICopilot significantly outperforms existing baselines in both automatic evaluation metrics and human evaluations. Specifically, statistical analysis reveals that the majority of human annotators prefer the webpages generated by ours over those produced by GPT-4V.[1]

## 1 Introduction

Translating webpage designs directly into code significantly streamlines the front-end development process, reducing both time and manual effort. This automation not only enhances developer productivity but also minimizes the potential for human error in code generation. However, earlier works were limited in scope, primarily relying on small deep-learning models to handle relatively simple *User Interface* (UI) design tasks. For instance, one of the earliest studies, pix2code [3] trained a model combining CNN [17] and LSTM [13] on a synthetic dataset to generate *Domain-Specific Languages* (DSL) code from simple UI images, which could be compiled into several front-end languages, including HTML for webpages. Sketch2code [33] explored generating webpage code from hand-drawn design sketches, investigating both deep learning and computer vision-based approaches.

With the rapid advancement of *Multimodal Large Language Model* (MLLM) , generating high-quality UI code from webpage designs (screenshots) has become increasingly feasible. Several studies have emerged in this field, focusing either on enhancing the code generation capabilities of MLLMs [10, 16] or evaluating their performance [10, 11, 36]. For instance, WebSight [16] introduced a training dataset generated by an LLM, while Design2Code [36] provided a curated test dataset of 485 samples along with an automatic metric to assess the similarity between generated webpages and their original designs. Vision2UI [10] contributed a real-world, large-scale webpage generation dataset for both training and evaluation. Similarly, Web2Code [49] presents a large-scale synthesized dataset along with an MLLM-based evaluation framework. When appropriate data are available, these works fine-tune MLLMs using pairs of webpage screenshots and their corresponding code. The resulting fine-tuned model operates in a one-step end-to-end approach, where it takes a screenshot as input and directly generates the corresponding code as output.

**Challenges and Motivation.** Despite the promising performance achieved by these one-step approaches, we are still far from fully automating UI synthesis for real-world webpages. As highlighted in [36], the complexity of code generation increases significantly as the total number of *Hypertext Markup Language* (HTML) tags, the diversity of unique tags, and the depth of the *Document Object Model* (DOM) tree grow. Existing MLLMs often experience a notable decrease in performance and efficiency when faced with real-world webpage designs that involve complex structures and a higher number of unique HTML tags [36]. Concretely, two primary challenges exist for the one-step generation approach.

**C1: The substantial length of the code that needs to be generated.** Common code generation tasks typically involve generating short code snippets or function implementations containing fewer than a few hundred tokens. In contrast, real-world webpages include not only HTML but also complex *Cascading Style Sheets* (CSS), significantly increasing the overall code length. For example, webpages in the Common Crawl dataset often contain tens of thousands of tokens, and even after extensive cleansing, they still average over 5,000 tokens [10]. This far exceeds the context window of most large models, posing significant challenges for both training and inference. Consequently, webpage generation is more akin to project-level development than simply generating isolated code snippets in one step.

**C2: The complexity of generating deeply nested structures.** Webpages are typically composed of multiple layers of nested elements, which makes generating these intricate structures from high-resolution design diagrams particularly challenging. Previous studies, such as [10, 25], have shown that even GPT-4V struggles to accurately capture structural information when evaluated on the pix2code test dataset, one of the simplest benchmarks for code generation from webpage designs.

---

[1] All the materials, including the source code, dataset, are available at: https://github.com/anonymouscodeeee/repo1.

**Our Work.** To mitigate these challenges, this paper puts forward UICOPILOT, a novel approach to automating UI synthesis via hierarchical code generation from webpage designs (screenshots). To alleviate the complexity of generating lengthy code, we decouple the process into two stages: generating the coarse HTML hierarchical structure and then producing fine-grained code. Concretely, we first introduce and train a *Vision Transformer* [8] (ViT)-based structure model to predict a coarse DOM tree containing only node types, hierarchy, and bounding boxes (BBoxes). To streamline the training process, we implement a BBox-based pruning strategy to simplify the dataset and reduce prediction complexity by removing nodes with areas below a fixed threshold, retaining only node type and hierarchy information. Using the predicted BBoxes of the coarse DOM tree's leaf nodes, the original design image is segmented into subregion images. These subregion images are sequentially fed into a code agent to generate the corresponding HTML/CSS code, which is then embedded back into the DOM tree. Finally, the DOM tree and design image are fed into the code agent once more to generate styles and attributes for the non-leaf nodes, while refining the global code.

To assess the effectiveness of UICOPILOT, we conduct experiments using the real-world dataset Vision2UI. We compare our results against several state-of-the-art baselines using three visual metrics. The experimental results show that UICOPILOT significantly outperforms other baselines. Notably, the visual score of GPT-4V improves by 23%, 27%, and 48% when integrated into UICOPILOT across three test datasets of increasing complexity, demonstrating that UICOPILOT is particularly effective in handling complex webpage generation. Additionally, we present several webpages generated by our method and GPT-4V in randomized order to human annotators for evaluation. The results reveal that in over 60% of the cases, annotators prefer the webpages generated by our method, providing strong evidence for the effectiveness of our hierarchical code generation approach.

The primary contributions of this paper are as follows:

- We propose a novel approach UICOPILOT that decouples the generation of hierarchical structure and fine-grained code. To the best of our knowledge, we are the first to break the limitations of one-stage webpage code generation by addressing the challenges of lengthy code generation and complex nested structures in real-world webpage Synthesis.
- We perform extensive experiments to evaluate the performance of UICOPILOT on the real-word Vision2UI dataset, and compare it with several state-of-the-art baselines. The results of automatic metrics and human evaluation show that UICOPILOT consistently outperforms other baselines.

## 2 Preliminaries

Before introducing our framework, we first discuss the inherent hierarchical structure of webpages and how MLLMs can be leveraged to generate code from webpage' design image.

### 2.1 Hierarchy Structure of Webpage Designs

Common webpage code is composed of three main components: *Hypertext Markup Language* (HTML), *Cascading Style Sheets* (CSS), and *JavaScript* (JS). HTML defines the elements on the page, their

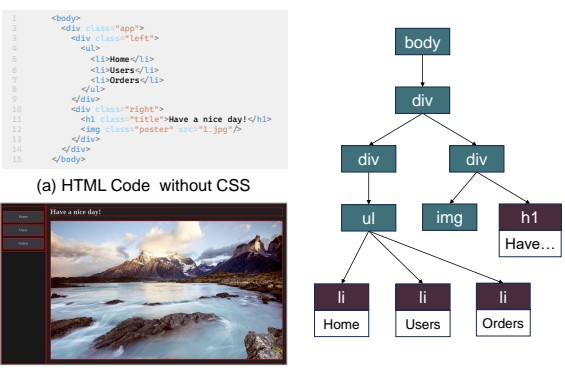

(a) HTML Code without CSS

(b) BBox

(c) HTML DOM Tree

**Figure 1: The structures of a webpage' design image.**

hierarchical structure, and their attributes. CSS is responsible for styling these elements, including properties such as size, color, and fonts. JavaScript typically manages the interactive functionality of the webpage, enabling dynamic behavior and user interactions. In our work, we focus on generating static webpages, so JavaScript code is excluded. The structural information in a UI design diagram primarily consists of the hierarchical relationships between elements, as well as their size and location. This information is crucial to the quality of webpage code generation and the accuracy of the final rendering. First, webpage code is organized in a hierarchical, nested structure, represented by the **HTML DOM Tree** (Figure1(c)), which serves as the skeleton of the webpage. Second, the size and location of UI elements are captured by **bounding boxes (BBox**, as shown in Figure 1(b)), which defines the primary layout structure. Our goal is to prioritize capturing the hierarchical structure of a webpage in order to generate code that closely aligns with the original design. Specifically, we aim to develop a webpage generation framework that accurately represents both the webpage's HTML DOM Tree and BBox information , ensuring that the resulting code reflects the design's structural and visual integrity.

### 2.2 Image to Code by MLLMs

MLLMs are designed to understand and generate multimodal content, such as text, images, and audio, by leveraging LLMs. Compared to unimodal models, MLLMs benefit from cross-modal knowledge transfer and understanding, showing great potential in tasks like image captioning, visual question answering, autonomous driving, and speech recognition. A typical MLLM consists of three key components: a modality encoder, a learnable connector, and an LLM. The modality encoder, often pre-trained on large-scale datasets (e.g., image-text pairs), enhances alignment with textual information. The learnable connector further aligns the different modalities during training. Rather than being trained from scratch, pre-trained LLMs are often utilized to improve efficiency and performance. The raw multimodal data is processed through the modality encoder, connector, and LLM to generate task-relevant text.

In our scenario, the primary objective is to generate code from images. We introduce two MLLMs: Pix2Struct-1.3B [18], which serves as the structure model for hierarchical structure prediction, and GPT-4V [28], which functions as the code agent. Common image encoders typically use either fixed-resolution or patch-based

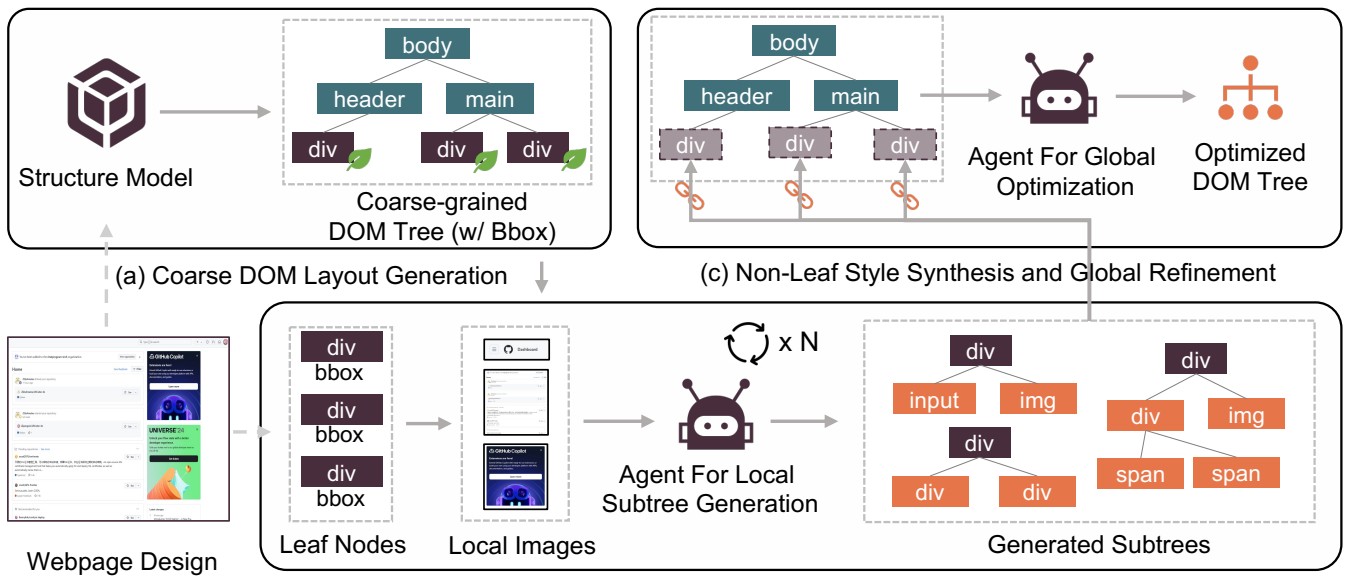

**Figure 2: Overview of the framework, including model training and inference.**

approaches. The fixed-resolution method scales and crops input images into a fixed-size matrix before feeding them into the encoder. In contrast, the patch-based approach, such as the Vision Transformer (ViT), scales the image to a specific ratio and divides it into fixed-size patches, which are then processed similarly to text tokens. We chose Pix2Struct as our base MLLM, which also uses an image encoder based on ViT but employs an aspect-ratio-preserving scaling strategy. This makes it more robust to extreme aspect ratios and adaptable to varying sequence lengths and resolutions. Given a UI image, after scaling, it is split into fixed-size patches. These patches are embedded and processed by the ViT encoder, which employs the self-attention mechanism to capture relationships between patches and understand the overall spatial arrangement of the UI components. The transformer-based text decoder of Pix2Struct then takes the encoded representation of the image and generates the corresponding output, e.g., UI component expressions, functions, and locations. Although Pix2Struct could generate simplified HTML from masked website screenshots, it is primarily designed for handling structured UI representations and still struggles with the full complexity of real-world webpages [18].

## 3  UICopilot

### 3.1  Task Description

In the task of webpage code generation, when given a high-resolution design image (such as a screenshot of an existing webpage), UICopilot should be able to generate the corresponding HTML and CSS code. After rendering, the generated webpage should closely align with the input image, particularly in terms of structure, style, and content. The generation of structure—specifically determining the type, size, and position of page elements—is critical to shaping the

webpage's overall visual appearance. As a result, a key focus of this work is the precise generation of well-formed webpage structures.

### 3.2  Overview

As shown in Figure 2, we decouple the webpage generation process into two stages: coarse DOM layout generation (Figure 2(a)) and fine-grained code synthesis (Figure 2(b&c)). In the first stage (See Figure 2(a)), we leverage a model to predict the webpage's coarse-grained HTML DOM tree and its BBoxes. This model is implemented using a ViT encoder and a transformer-based decoder and is trained with a BBox-based pruning strategy to reduce noise in the dataset, resulting in enhanced model accuracy and accelerated convergence. In the fine-grained code synthesis stage, our framework incorporates an existing MLLM as a code agent to generate detailed local code and styles. The local images of the coarse DOM tree's leaf nodes are fed into the code agent to generate the corresponding local code, which is then linked back to the tree (Figure 2(b)). Afterward, the integrated code is fed back into the code agent to supplement the styles and attributes of the non-leaf nodes in the coarse DOM tree while refining the global code (Figure 2(c)). Through this hierarchical code generation approach, we ultimately produce a complete webpage that effectively captures the structural information while maintaining detailed accuracy.

This generation method is based on the assumption that "*when the UI elements and structure in an image are sufficiently simple (such as local region images in design layouts), MLLMs can effectively generate webpage code that captures both structure and detail.*" By decoupling the generation of the DOM tree and positional information from the detailed local code, this approach reduces the burden on large models to generate excessively long code. Prioritizing the DOM tree and positional information also ensures a more accurate

capture of the original webpage's structural elements, addressing the challenges outlined in the introduction.

## 3.3 Coarse DOM Layout Generation

To generate webpage structure information more accurately and efficiently, we introduce and train a structure model specifically designed to predict a coarse-grained version of the HTML DOM tree, capturing only the node types, hierarchy, and BBoxes of the nodes. Additionally, we carefully designed the data processing and training procedures to optimize performance.

**Structure Model.** In our application scenario, high-resolution webpage designs are typically presented in various resolutions. Cropping or resizing the images can easily lead to the loss of crucial information, which is not conducive to the final generation of UI code. Given that the ViT processes images into multiple patches to maximize the preservation of information from the original image, we adopt the Pix2Struct model which is based on ViT, as our fundamental structure model. Pix2Struct distinguishes itself by its robustness towards extreme aspect ra- tios and on-the-fly changes to the sequence length and resolution

The structure model's task is framed as a sequence generation problem using a next-token prediction mechanism. Given an input image $I$, a high-resolution webpage screenshot, the model generates a structured representation of the webpage's DOM tree in JSON-formatted text. This JSON text encodes the DOM tree as nested HTML code with accompanying BBox attributes. The process begins by dividing the input image $I$ into fixed-size patches, represented as $\{P_1, P_2, \ldots, P_N\}$. These patches are then passed through the ViT encoder, denoted as $\text{Enc}_\theta$, which converts them into hidden state vectors $\{h_1, h_2, \ldots, h_N\}$:

$$\{h_1, h_2, \ldots, h_N\} = \text{Enc}_\theta(\{P_1, P_2, \ldots, P_N\})$$

These hidden state vectors are subsequently fed into the decoder, $\text{Dec}_\phi$, a conventional transformer-based text decoder, to predict the JSON representation of the DOM tree and their corresponding BBoxes. Leveraging the encoder's hidden states, the decoder generates the JSON text one token at a time. At each time step $t$, the decoder generates a probability distribution for the next token $y_t$, conditioned on all previous tokens $y_1, y_2, \ldots, y_{t-1}$. This prediction is computed using a softmax function applied to the decoder's hidden state $h_t$, which integrates information from both the encoder's hidden states $\{h_1, h_2, \ldots, h_N\}$ and the previously generated tokens, projected by a learned weight matrix $W$:

$$P(y_t \mid y_1, y_2, \ldots, y_{t-1}) = \text{softmax}(W \cdot h_t)$$

**BBox-based Data Pruning.** We use the Vision2UI dataset, comprising real webpage screenshots and corresponding code, to train our model. However, throughout our investigation, we find that the BBoxes of HTML elements in real-world webpages often contain a significant amount of noise, such as empty or very small elements, as well as elements representing visually hidden parts of the webpages or parts that are not displayed correctly (as illustrated in Figure 3(a)). This noise not only severely reduces the model's learning efficiency but also contributes little to the overall structural information of webpages. Therefore, we applied several heuristic rules to prune the original BBoxes along with their corresponding webpage elements: 1) We first remove BBoxes smaller than 3% of

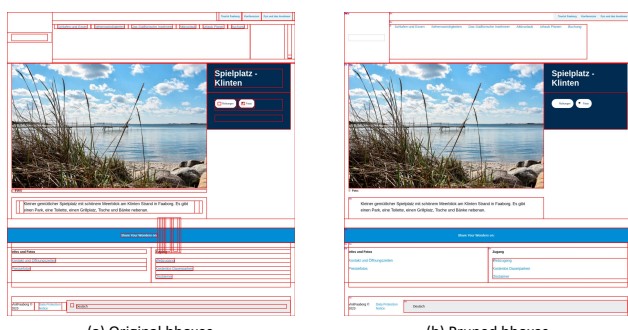

(a) Original bboxes                (b) Pruned bboxes

**Figure 3: BBox-based data pruning.**

the total area, along with all their child nodes. We reasonably assume that for simple content within small areas, the MLLM can efficiently generate the corresponding high-quality structure and style code. 2) We eliminate BBoxes that contain only a single type of pixel, along with all their child nodes. These BBoxes are usually empty and contain no actual web elements. 3) We discard webpage samples with fewer than 10 total BBoxes on the entire page. An insufficient number of BBoxes typically indicates potential errors during processing.

Figure 3(b) presents an example of the pruned BBoxes, where most of the noise has been eliminated, revealing a clearer and simplified hierarchical structure. During model training, we found that this pruning method significantly accelerates model convergence while preserving as much of the original hierarchical structure of the webpage as possible. In the second phase, this streamlined BBox information greatly improves both the generation quality and efficiency of the MLLM.

During the inference process, we apply pruning rules to the results generated by the structure model, primarily based on minimum area (`min_area`) of BBoxes and maximum depth (`max_depth`) of DOM tree. This ensures that the predicted DOM tree and BBoxes are concise and appropriate, setting a better stage for fine-grained code generation. In the experimental section, we conducted controlled experiments to specifically study the impact of these two parameters.

**Model Training.** To train the structure model, we formulate the task as a conditional sequence generation problem. The primary objective is to optimize the model parameters to maximize the likelihood $P(J \mid I)$ of the correct JSON text $J$ given the input image $I$. This is achieved by minimizing the negative log-likelihood of the correct tokens:

$$\mathcal{L}(\theta) = -\sum_{t=1}^{T} \log P(j_t \mid j_1, j_2, \ldots, j_{t-1}, I; \theta)$$

where $\theta$ represents the model parameters. The probability $P(j_t \mid j_1, j_2, \ldots, j_{t-1}, I; \theta)$ is computed using the softmax function over the decoder's output logits:

$$P(j_t \mid j_1, j_2, \ldots, j_{t-1}, I; \theta) = \frac{\exp(z_{t,j_t})}{\sum_{k=1}^{V} \exp(z_{t,k})}$$

where $z_{t,k}$ is the logit for token $k$ at time step $t$, and $V$ is the size of the vocabulary.

You are a skilled web developer specializing in building webpages.
**CONTEXT**
I am working on a project that involves converting webpage design images into functional code. Your task is to generate the corresponding HTML code for specific segments of the webpage based on the provided module names and images.
**OBJECTIVE**
Generate the partial HTML code based on the input webpage image and the initial node type.
**RESPONSE**
Provide the HTML code necessary to implement the module's functionality, including inline CSS.
**INITIALIZE**
In the upcoming messages, I will send you the webpage image and module name. Upon receiving them, please follow the above instructions to generate the corresponding HTML code (the highest-level node in the generated HTML tree should match the given initial node type).

**Figure 4: Prompts for leaf node HTML/CSS generation.**

You are a skilled web developer specializing in building webpages.
**CONTEXT**
I am working on a project that converts webpage design images into functional code. Your task is to adjust and optimize the styles of the already generated webpage code, based on the webpage image, without altering the original DOM tree structure of the code.
**OBJECTIVE**
Adjust and optimize the styles according to the input webpage image and original HTML code.
Do not modify the node types or hierarchical structure of the DOM tree in the original code!!!
Retain the original DOM tree nodes exactly as they are.
Do not change the image src attributes in the code.
**RESPONSE**
Adjust and optimize the CSS styles without altering the DOM tree structure in the original HTML code.
**INITIALIZE**
In the upcoming messages, I will send you the webpage image and the existing webpage code. Upon receiving them, please follow the above instructions to adjust and optimize the CSS styles in the original HTML code.

**Figure 5: Prompts for non-leaf style synthesis and global refinement.**

## 3.4 Fine-Grained Code Synthesis

While the generated coarse DOM tree contains the node types and hierarchy, it omits two crucial aspects: (1) The structural and style code for the local regions of the leaf nodes. Since the structure model is designed to predict a coarse-grained DOM tree, and this prediction is further refined based on *min_area* and *max_depth*, the leaf nodes in the DOM tree actually serve as parent nodes for sub-regions that still contain sub-trees requiring further prediction. (2) The attributes and styles of the non-leaf nodes within the coarse DOM tree. To address these two missing elements, we have devised the following two steps to complete the generation process.

**HTML/CSS Generation of Leaf Node.** In practice, we employ GPT-4V as our local code generation agent, leveraging its flexibility and robust code generation capabilities. Since the BBoxes contain the size and position information of the nodes, we use the BBoxes of the leaf nodes in the predicted coarse DOM tree to crop images of the corresponding regions from the original image. These segmented images are then fed individually into the code agent to predict the corresponding HTML/CSS code for each subregion. After obtaining the HTML/CSS code for the leaf nodes, we embed them back into the corresponding leaf nodes of the coarse DOM tree. By doing so, we isolate the visual content associated with each leaf node, allowing the agent to generate accurate HTML/CSS code

**Table 1: A statistical comparison between both WebSight and Vision2UI. The statistical data of the two is referred to [36].**

|  | WebSight | Vision2UI |
| --- | --- | --- |
| Purpose | Training | Training&Tesing |
| Source | Synthetic | Real-World (Common Crawl) |
| Size | 0.8M | 3.1M |
| Avg. Len (tokens) | 647±216 | 4661±2006 |
| Avg. Tags | 19±8 | 188±80 |
| Avg. DOM Depth | 5±1 | 15±5 |
| Avg. Unique Tags | 10±3 | 24±6 |

for these specific regions one by one, significantly alleviating the burden of generating lengthy code and improving the generation quality. We have carefully designed a prompt that instructs the agent to generate the corresponding HTML/CSS code based on the input subregion image and the parent node type, as shown in Figure 4.

**Non-Leaf Style Synthesis and Global Refinement.** At this stage, we feed the code obtained from the previous step along with the entire design image into the agent, instructing it to supplement the coarse DOM tree's non-leaf nodes with styles and other attributes. Specifically, the agent analyzes the global layout and visual elements of the full design image to infer styling information such as fonts, colors, margins, paddings, and other CSS properties for the non-leaf nodes. This ensures that the final code not only reflects the correct structural hierarchy but also accurately represents the visual aesthetics of the original design. Additionally, we require the agent to refine the global code while ensuring that the input DOM tree remains unchanged. The instructions used in this process are presented in Figure 5.

## 4 Experiments and Analysis

### 4.1 Datasets

**Training Dataset.** We utilized two datasets, WebSight and Vision2UI, for training, each demonstrating distinct characteristics across various metrics. The WebSight v0.1 dataset, comprising approximately 0.8 million data entries, is synthesized using two language models (LM): a smaller LM first generates themes and designs, which are then fed into an LLM trained on code bases to produce the final HTML code using carefully crafted prompts. The Vision2UI dataset, consisting of about 3.1 million data entries, is derived from the real-world Common Crawl dataset[2]: the authors filter the original dataset based on length and other criteria to eliminate noise, such as invisible elements and comments, and subsequently employ a scorer trained on a manually annotated dataset to further enhance data quality. Compared to the WebSight dataset, Vision2UI's data is more complex, possesses a richer variety of styles and tag types, and is significantly longer, making it closer to real-world HTML code. Furthermore, Vision2UI provides the page's BBox information directly, which is essential for the training of our structure model. For the WebSight dataset, we also extracted the BBox information prior to training.

**Test Dataset.** We evaluate our framework on the Vision2UI test datasets. Vision2UI test datasets are composed of three subsets:

---

[2]https://data.commoncrawl.org/

Vision2UI-short, Vision2UI-mid, and Vision2UI-long. These subsets are obtained by dividing according to the length range of the ground-truth HTML code, combined with manual selection. The length ranges of the ground-truth HTML code for these three subsets are [551, 2045], [2052,4085], and [4098,10990] respectively. Each subset contains 256 samples.

## 4.2 Evaluation Metrics

**CLIP Similarity.** CLIP is a multi-modality model trained by contrastive objective in a dataset of millions of internet text-image pairs, learning to align images and their textual descriptions in a common representation space. The latent vectors produced by CLIP model encode semantic information of the input. Images that are similar will have similar context and thus similar vectors. Hence, the cosine similarity of these vectors, which measures the cosine of the angle between them, effectively captures the degree of similarity between the images. It is caculated by $cosine(CLIP(image_i), CLIP(image_j))$.

**SSIM.** The Structural Similarity Index (SSIM) [46] considers changes in structural, luminance, and contrast information in the images. SSIM is a more comprehensive measure than traditional methods like MSE or PSNR, as it simulates the human visual perception system, which is highly adapted for extracting structural information.

**Visual Score.** This is a concept proposed by Design2code [36], and is utilized to measure the matching degree of low-level elements in terms of appearance. These scores primarily calculate the match ratio between the reference and candidate blocks, as well as the similarity at the block level in terms of color, text, and position. Since there are numerous formulas involved, it is a little trivial to list them all here.

## 4.3 Baselines

The baselines in our work can be categorized into finetune-based MLLMs that are specialized for webpage generation and general-purpose MLLMs. The specialized MLLMs include:

- **WebSight VLM-8B.** Hugging Face's WebSight utilizes its training dataset and the DoRA [22] mechanism to finetune a VLM that has been pre-trained on image/text pairs.
- **Design2Code-18B.** Stanford's Design2Code is also fine-tuned on the WebSight dataset. However, it adopts CogAgent as its base model and utilizes LoRA [14] as the finetuning method to accelerate the training process.

The general-purpose MLLMs (prompt-based) include:

- **CogAgent-Chat-18B.** CogAgent-Chat-18B is a general MLLM that supports both low- and high-resolution images. Notably, it performs well on webpage navigation, requiring only screenshots. We input the screenshot and a simple prompt *Write an HTML code* to generate the webpage, similar to Design2Code.
- **GPT-4V.** GPT-4V, an advanced AI model, demonstrates remarkable capabilities in image comprehension. It also possesses the unique ability to generate code from images. We referred to the prompt of the well-known open-source project *screenshot-to-code* [3] on GitHub, with slight modifications.
- **LLaVA-v1.5-7B [21].** LLaVA-v1.5-7B is an end-to-end trained large multimodal model that connects a vision encoder and an

LLM for general-purpose visual and language understanding. We use the same prompt as GPT-4V to generate webpages from images.

Although previous work [36] suggests that multi-round generation methods (e.g., self-revision) may outperform one-pass approaches, important baselines such as Design2Code-18B and WebSight VLM-8B are models obtained through fine-tuning and only support one-pass generation. Therefore, to ensure a fair comparison, all baselines employ the one-pass generation method.

## 4.4 The effectiveness of UICOPILOT

**Overall performance.** Table 2 presents the performance breakdown of UICOPILOT and the baselines on the Vision2UI test datasets. From the table, we observe that our method's visual score and CLIP similarity significantly outperform all the baselines across the three test datasets. This leading performance in the CLIP metric indicates that the webpages generated by UICOPILOT are more visually similar to the original ones in terms of overall appearance and features. In terms of SSIM, LLaVA holds a slight advantage over the other models, suggesting that the webpages it generates may be more closely aligned with the originals concerning visual luminance, contrast, and structure. However, the differences among all models in this metric are not significant. Our UICOPILOT also performs well on this metric compared to the other models.

Notably, while our approach utilizes GPT-4V for generating fine-grained code, UICOPILOT still outperforms GPT-4V in terms of visual score by 23%, 27%, and 48% across the three test datasets, respectively. Additionally, our approach exhibits significantly lower variance in visual scores, indicating higher stability across all test samples. Moreover, it's worth noting that as the length of the sample webpages in the test datasets increases, the advantage of UICOPILOT over GPT-4V grows substantially. This further demonstrates that UICOPILOT's architecture—which focuses on decoupling the structure parsing from the fine-grained code generation processes—reduces complexity and results in more accurate HTML/CSS generation from design images.

**Visual Score breakdown.** The `visual score` is a composite metric consisting of five sub-indicators: block-level color, text, position, text color, and CLIP similarity information. In Figure 6, we provide a detailed analysis of these sub-indicators. To further explore the contribution of the refinement process in our framework, we introduce UICOPILOT without code refinement (denoted as UICOPILOT w/o opt) for comparison. As shown in Figure 6, UICOPILOT w/o opt already outperforms GPT-4V in terms of CLIP similarity, text, position, and text color. This demonstrates that the input coarse-grained HTML DOM tree and BBoxes effectively enhance GPT-4V's generation capability, particularly evident in the significant improvement in the text color indicator. However, we observe that UICOPILOT w/o refinement performs worse than GPT-4V in the block-level color indicator. Our investigation suggests that strictly adhering to the coarse-grained HTML DOM tree and BBoxes without refinement can lead to missing styles or errors in the upper layers of nodes, negatively impacting the block-level color. After incorporating the refinement process, as seen in Figure 6, there is a significant improvement in the block-level color indicator, highlighting the effectiveness of the refinement process.

---

[3]https://github.com/abi/screenshot-to-code

Table 2: The performance breakdown on the visual metrics.

| Model | Vision2UI-Short | | | Vision2UI-Mid | | | Vision2UI-Long | | |
|---|---|---|---|---|---|---|---|---|---|
| | Visual Score | CLIP | SSIM | Visual Score | CLIP | SSIM | Visual Score | CLIP | SSIM |
| WebSight VLM-7B | 0.57 (±0.24) | 0.69 (±0.12) | 0.62 (±0.17) | 0.52 (±0.23) | 0.67 (±0.11) | 0.59 (±0.16) | 0.48 (±0.27) | 0.64 (±0.11) | 0.61 (±0.15) |
| Design2Code | 0.75 (±0.14) | 0.68 (±0.10) | 0.58 (±0.15) | 0.69 (±0.23) | 0.70 (±0.10) | 0.56 (±0.14) | 0.61 (±0.28) | 0.68 (±0.10) | 0.61 (±0.11) |
| CogAgent-Chat | 0.46 (±0.31) | 0.68 (±0.11) | 0.59 (±0.15) | 0.40 (±0.31) | 0.66 (±0.10) | 0.58 (±0.14) | 0.39 (±0.30) | 0.65 (±0.10) | 0.60 (±0.13) |
| LLaVA-v1.5-7B | 0.43 (±0.27) | 0.63 (±0.11) | **0.65 (±0.17)** | 0.21 (±0.28) | 0.63 (±0.10) | **0.65 (±0.14)** | 0.19 (±0.27) | 0.61 (±0.10) | **0.66 (±0.12)** |
| GPT-4V | 0.68 (±0.32) | 0.74 (±0.11) | 0.61 (±0.14) | 0.65 (±0.33) | 0.71 (±0.10) | 0.55 (±0.12) | 0.62 (±0.35) | 0.67 (±0.11) | 0.57 (±0.11) |
| UICopilot | **0.84 (±0.18)** | **0.77 (±0.11)** | 0.60 (±0.13) | **0.83 (±0.17)** | **0.77 (±0.10)** | 0.57 (±0.12) | **0.78 (±0.24)** | **0.74 (±0.10)** | 0.60 (±0.11) |

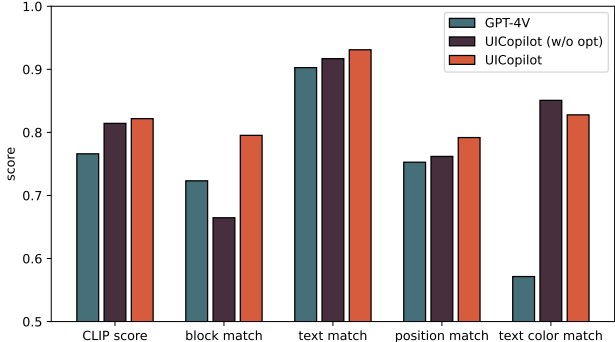

Figure 6: Detailed sub-indicators of Visual Score

## 4.5 The influence of `min_area` and `max_depth`

the structure model, based on two parameters: minimum area of BBox (`min_area`) and maximum depth of DOM tree (`max_depth`). To determine the optimal values for these parameters, we conducted a grid search exploring different combinations. However, since performing inference on the entire test dataset is time-consuming, we optimized the grid search process by constraining the parameters to practical ranges based on empirical experience: [10%, 20%, 30%, No limit] for `min_area` and [4, 5, 6, No limit] for `max_depth`. We focused on the metric with the greatest relative improvement in our method, the `visual score`, for further experiments. By analyzing all sub-metrics of the visual score under different combinations of `min_area` and `max_depth`, we obtained the results presented in Figure 7. Our analysis revealed that nearly all sub-metrics exhibit a consistent and clear linear trend: as both `max_depth` and `min_area` decrease, the metrics improve, indicating a visual enhancement in the generated results. Specifically, we found that setting `max_depth` to 4 and `min_area` to 10% yields a balanced local optimum in terms of visual matching.

## 4.6 Human Evaluation

From Table 2, we observe minimal differences between UICopilot and GPT-4V in terms of CLIP similarity and SSIM, both of which primarily assess overall image similarity. To address the question "*Which model generates webpages closer to the original design?*", we conducted a human evaluation experiment. In this experiment, we present pairs of webpages generated by UICopilot and GPT-4V to six annotators, shuffling the samples in each pair to eliminate bias. The annotators are asked to select the webpage they felt is most aligned with the original design. As illustrated in Figure 8, over 60% of the selections preferred UICopilot, providing strong evidence of its effectiveness in producing webpages that more closely match the original designs.

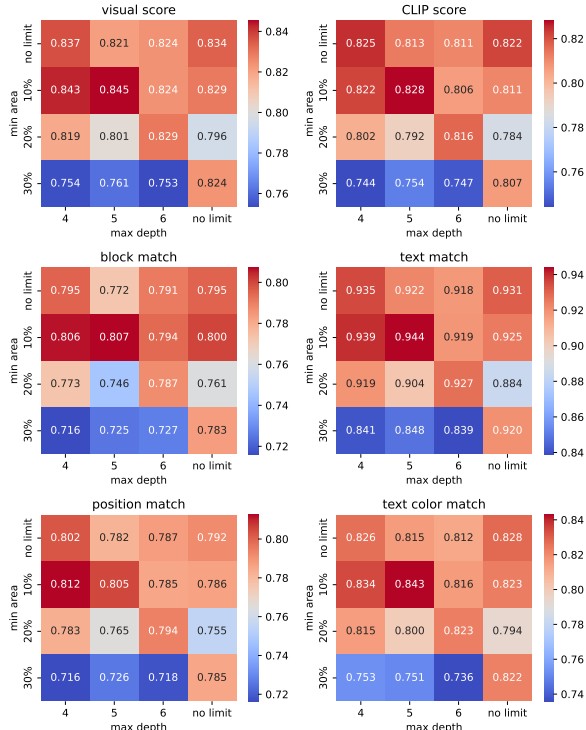

Figure 7: Results of the parameter grid search.

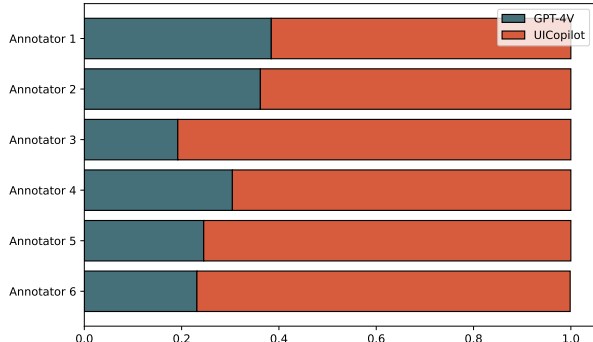

Figure 8: The preference of human annotators.

## 4.7 Case Study

To better illustrate the advantages of UICopilot in webpage generation, we select a representative example shown in Figure 9. The images from left to right depict the original webpage, the webpage generated by GPT-4V, and the one generated by UICopilot. As observed, both GPT-4V and UICopilot capture the footer, body,

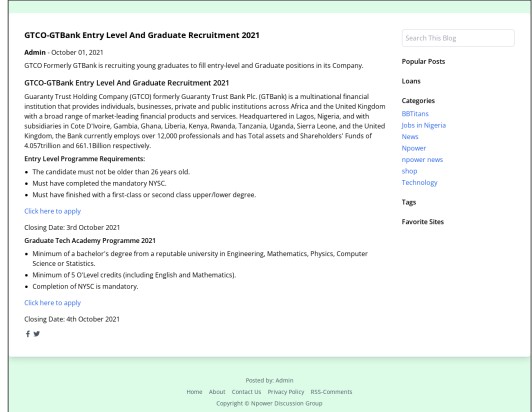
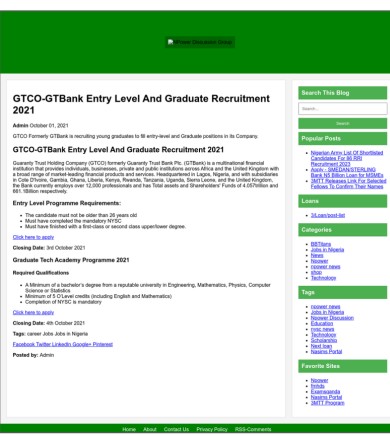

(a) Original  (b) GPT-4V  (c) UICopilot

**Figure 9: A representative example of generated webpages.**

and header of the webpage, and the text content is fairly close to the original. However, in terms of structural details, GPT-4V fails to replicate the various small blocks on the right side of the original image, opting instead for a simplified list layout. In contrast, UICopilot successfully captures these structural details. This demonstrates that decomposing webpage code synthesis into a coarse-grained DOM tree and localized fine-grained code generation significantly enhances the ability to capture structural elements. Nevertheless, both models exhibit shortcomings in finer details: some text is missing, colors do not completely match the original, and the sizes of certain sections differ from the original layout. These deficiencies suggest that post-generation edits and localized fixes could considerably enhance the quality of the generated webpages, indicating an area for further exploration.

## 5 Related Work

**Code Generation.** Recently, notable advancements have been made in code generation through various pre-trained code language models. For instance, CodeGPT [23], a Transformer-based model following a similar architecture to GPT-2 [30], was trained on a corpus tailored for program synthesis. Another model, CodeT5 [45], built upon T5 [31], was pre-trained across eight programming languages and integrated an identifier-aware objective during its pre-training phase. Additionally, Codex [6], a GPT-based model trained on code from GitHub, has notably served as the foundational framework for Copilot [1]. Moreover, AlphaCode [20] stands out as a code generation system designed to produce unique solutions for intricate problems requiring deep cognitive engagement. More recently, the landscape of code generation has been significantly influenced by large language models (LLMs) such as CodeGen [26], CodeT5+[44], InCoder[9], GPT-3.5 [27], StarCoder [19], Code Llama [35], and WizardCoder [24].

**Image Representation Learning.** To obtain better image representations, early works used Variational Autoencoders (VAEs) to generate latent vectors [15, 41], while others employed contrastive learning to derive image encoders from large datasets [7]. Unlike conventional CNN-based models with attention mechanisms [5, 43], the Vision Transformer (ViT) [2] breaks images into fixed-size

patches and processes them directly with transformers. To reduce the computational burden of diffusion models (DMs) [12] in pixel space, researchers proposed training DMs in the latent space of advanced pre-trained autoencoders [34], transforming DMs into robust generators for various conditioning inputs via cross-attention layers. Recently, notable works have explored large models for image understanding, including SDXL [29], VideoLDM [4], and others [38, 39, 42, 48].

**Image to Code.** With the rapid advancement of LLMs in recent years, many researchers have focused on generating code from images. Wu et al. [47] formulated the problem of *screen parsing*, predicting UI hierarchy graphs from screenshots using Faster-RCNN [32] to encode screenshot images and an LSTM attention mechanism to construct graph codes and edges. Pix2Struct [18], pre-trained to predict simplified HTML from masked website screenshots, significantly improved visual language understanding on nine tasks across four domains. To address the computational burden and non-differentiable issues of website rendering, Soselia et al. [37] applied reinforcement learning to fine-tune a vision-code Transformer (ViCT)—comprising a visual ViT [8] and a GPT-2/Llama-based code decoder [30, 40]—by minimizing discrepancies between the original and generated webpage without rendering.

## 6 Conclusion

In this work, to address the challenge of generating lengthy webpage code and to better capture the structural information in webpage design images, we decoupled the generation procedure into coarse DOM tree generation and fine-grained code synthesis. We introduced a ViT-based structure model to predict the coarse DOM tree and trained the model along with a BBox-based data pruning strategy. With the coarse DOM trees and BBoxes predicted by the structure model, we decomposed the process of generating code from the original image into generating code snippets of segmented subregion images and assembling them together, which avoids generating long code in a single step. The experimental results and feedback from human annotators demonstrate the effectiveness of the proposed framework.

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
