# OpenReview forum: "UICopilot: Automating UI Synthesis via Hierarchical Code Generation from Webpage Designs"
_ACM.org/TheWebConf/2025/Conference — WWW 2025 Oral_

### Official Review · Reviewer_WJLJ · 2024-11-16

**Novelty:** 5
**Technical Quality:** 5

**Review:**

#### **Quality**

The paper presents UICopilot, a framework for automated user interface (UI) synthesis that generates hierarchical HTML structures and CSS from webpage design images. The authors introduce a two-stage approach, decoupling coarse layout generation from detailed code synthesis. This work leverages the Vision2UI dataset to validate performance. The technical foundation is solid, and the method effectively mitigates challenges associated with lengthy and complex UI code generation.

#### **Clarity**

The presentation is generally clear, providing a well-structured breakdown of the framework's functionality and the experiments.


#### **Originality**

The originality of the UICopilot framework stands out, particularly through its **two-stage hierarchical approach**, which is the first to decouple UI synthesis tasks in this manner. By incorporating a coarse HTML structure generation stage followed by fine-grained code synthesis, the method addresses limitations of one-step models, presenting a substantial improvement in performance and structure.


#### **Significance**

UICopilot’s contributions are significant for the field, as it showcases notable advancements in automating UI synthesis, a task with extensive applications in web development and UI/UX design. The performance improvements over state-of-the-art models like **GPT-4V** and **Design2Code** suggest this method could set a new standard for UI code generation.

**Pros:**

* Proposes a novel hierarchical approach to UI synthesis that effectively manages complex webpage structures and lengthy code requirements.
* Empirical results on Vision2UI demonstrate UICopilot’s enhanced performance over baseline models in both visual metrics and human evaluations.

**Cons: **

The term “agent” is used ambiguously, with limited discussion on its autonomy or relation to AI agents. This lack of clarity may confuse readers regarding the agent’s role and level of independence within UICopilot.

**Questions:**

# **Questions**

**Q1. **UICopilot’s hierarchical design, including the Coarse DOM Layout Generation stage, is central to its approach. However, the paper does not assess whether this stage is essential. Analyzing the model’s performance without it—using GPT-4V directly on isolated BBox screenshots—would clarify the value of hierarchical guidance. This could reveal if the hierarchy significantly enhances layout accuracy and visual consistency or if similar outcomes could be achieved with a simpler approach.
**Could the authors provide analysis on UICopilot’s performance with and without the Coarse DOM Layout Generation? Specifically, does the hierarchical structure meaningfully improve visual consistency and layout accuracy when generating HTML/CSS from isolated BBox screenshots?

**Q2**. Could the authors clarify the use of “agent” in UICopilot? Specifically, does this “agent” possess any autonomy or decision-making abilities typical of AI agents, or is it simply a functional component within the hierarchical code generation framework?

**Reviewer Confidence:**

3: The reviewer is confident but not certain that the evaluation is correct

**Scope:**

4: The work is relevant to the Web and to the track, and is of broad interest to the community

---

### Official Review · Reviewer_kthb · 2024-11-28

**Novelty:** 5
**Technical Quality:** 5

**Review:**

The paper introduces UICopilot, a hierarchical approach for automating UI synthesis from webpage designs using Multimodal Large Language Models (MLLMs). The approach addresses the complexity of generating long and nested code by decoupling the process into two stages: generating a coarse HTML structure and producing fine-grained code. UICopilot's effectiveness is demonstrated through experiments on the Vision2UI dataset, where it significantly outperforms baseline models, including GPT-4V, in terms of both automatic metrics and human evaluations.

Strengths:
1. The hierarchical structure of UI code generation (coarse-to-fine) is a effective approach to solving the problem of long and complex webpage generation.
2. The paper conduct comprehensive experiments. The results show that UICopilot outperforms state-of-the-art baselines in both quantitative and qualitative evaluations, which demonstrates its potential for real-world applications.


Weaknesses:

1. While the decoupling approach is innovative, the overall concept of UI code generation using MLLMs has been explored in earlier works (e.g., Pix2Code, Sketch2Code). The novelty could be further emphasized.
2. The method, while effective, is somewhat complex, and the explanation could benefit from simplification, especially in the process of hierarchical code generation (Section 3).

**Questions:**

1. While the decoupling approach is innovative, the overall concept of UI code generation using MLLMs has been explored in earlier works (e.g., Pix2Code, Sketch2Code). The novelty could be further emphasized.
2. The method, while effective, is somewhat complex, and the explanation could benefit from simplification, especially in the process of hierarchical code generation (Section 3).

**Reviewer Confidence:**

3: The reviewer is confident but not certain that the evaluation is correct

**Scope:**

3: The work is somewhat relevant to the Web and to the track, and is of narrow interest to a sub-community

---

### Official Review · Reviewer_qWe9 · 2024-12-02

**Novelty:** 5
**Technical Quality:** 4

**Review:**

This paper presents a framework based on Multi-modal Large Language Model (MLLM), aimed at generating high-quality static web page code (HTML and CSS) from webpage screenshots. The framework enhances the accuracy and efficiency of webpage generation by combining visual and structural information. The overall quality of this work is relatively average; the content is clearly and logically articulated, with moderate originality. The results are relatively evident and contribute to the development of the field.
The merits of this paper include the following: Firstly, the decoupling approach proposed in this work improves the quality of nested webpage code generation. By comparing it with multiple SOTA baseline models, the paper demonstrates the superior performance of UICopilot on the real-world Vision2UI dataset, validating the effectiveness and practicality of this method in web page generation tasks. Additionally, the adoption of the BBox-based Data Pruning method partially addresses the issue of long and noisy real-world webpages, as discussed in the design2code work.
On the other hand, there are aspects of the paper that could be further improved. First, regarding the JSON text generated in the first stage, there is a lack of example demonstrations, since the output format is closely tied to the generation in the subsequent phase. Second, in the code refinement stage, the paper does not specify which model or Python library is used for webpage image cropping, nor does it address whether the algorithm's precision is sufficient or if errors may propagate. This needs further clarification. Third, regarding the model performance metrics, an in-depth analysis of the generally low SSIM scores would be beneficial to facilitate future targeted improvements. Fourth, the ablation study has insufficient data; Figure 6 only provides a breakdown of visual score without showcasing other indicators, such as SSIM. It would also be useful to display ablation results across other dimensions, such as using LLMs in both phases or removing pruning operations, to present a more comprehensive evaluation of the model's performance. Lastly, while the overall language in the paper is clear, there are some minor errors, such as the omission of a reference to Table 1.

**Questions:**

1. Regarding the JSON text generated in the first stage, there is a lack of example demonstrations.
2. In the code refinement stage, which model or Python library is used for webpage image cropping, whether the algorithm's accuracy is sufficient and whether it could lead to error propagation.
3. The ablation study has insufficient data; Figure 6 only provides a breakdown of visual score without showcasing other indicators, such as SSIM. And it would also be useful to display ablation results across other dimensions.
4. An in-depth analysis of the generally low SSIM scores would be beneficial to facilitate future targeted improvements.

**Reviewer Confidence:**

3: The reviewer is confident but not certain that the evaluation is correct

**Scope:**

4: The work is relevant to the Web and to the track, and is of broad interest to the community

---

### Official Review · Reviewer_oMnQ · 2024-12-03

**Novelty:** 5
**Technical Quality:** 5

**Review:**

This article introduces a method for generating HTML and CSS code from images, presenting a hierarchical approach to enhance code generation. Unlike prior works, the authors propose a two-step method designed to maintain performance even when generating substantially lengthy code.

The text is clear, well-written, and supported by results that demonstrate the potential of the proposed method. Below is an evaluation of its strengths and weaknesses:

Strengths:

Performance: The proposed approach outperforms baseline methods in most cases, demonstrating its effectiveness.

Robustness to Code Length: By grouping the test dataset based on code length, the authors show that their method maintains reasonable performance regardless of the code's length, outperforming the baseline in these scenarios.

Comparative Analysis: The authors provide They have done an analysis when their approach is better than GPT-4V

Weaknesses:

Human Evaluation: provide a more detailed explanation of the Human Evaluation process.

Choice of Baseline: The justification for selecting specific baseline methods from related works needs to be elaborated

Limitations and future works: the article lacks a discussion of the limitations and future work of their approach

**Questions:**

Q1: In the case study, why did the authors choose baseline GPT-4V over other approaches for comparison?

Q2: The human evaluation study lacks of a better explanation: For example, does it was used the whole test dataset? How the dataset was divided along the 6 annotators?

Q3: Could you discuss in greater detail the limitations of your approach and what future directions might help overcome these limitations?

**Reviewer Confidence:**

3: The reviewer is confident but not certain that the evaluation is correct

**Scope:**

3: The work is somewhat relevant to the Web and to the track, and is of narrow interest to a sub-community